

# Bethe-state counting and Witten index

**Hongfei Shu[1,2,3], Peng Zhao[1,4], Rui-Dong Zhu[5] and Hao Zou[1,2]**

**1** Beijing Institute of Mathematical Sciences and Applications, Beijing 101408, China
**2** Yau Mathematical Sciences Center, Tsinghua University, Beijing 100084, China
**3** School of Physics and Microelectronics, Zhengzhou University, Zhengzhou, Henan 450001, China
**4** Joint School of the National University of Singapore and Tianjin University, International Campus of Tianjin University, Fuzhou, 350207, China
**5** Institute for Advanced Study & School of Physical Science and Technology, Soochow University, Suzhou 215006, China

## Abstract

We count the physical Bethe states of quantum integrable models with twisted boundary conditions using the Witten index of 2d supersymmetric gauge theories. For multicomponent models solvable by the nested Bethe ansatz, the result is a novel restricted occupancy problem. For the SU(3) spin chain and the t-J model, we propose formulae for the solution count on singular loci in the space of twist parameters.

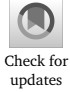

# 1 Introduction

The Heisenberg spin chain is the prototype of all quantum integrable models. The exact solution is given in terms of the Bethe ansatz equations (BAEs) [1]

$$\left(\frac{\lambda_k + i/2}{\lambda_k - i/2}\right)^L = \prod_{\substack{j=1 \\ j \neq k}}^{M} \frac{\lambda_k - \lambda_j + i}{\lambda_k - \lambda_j - i}. \tag{1}$$

The equations impose quantization conditions on the magnon rapidities $\{\lambda_k\}$ that define eigenstates with $M$ down spins and $L - M$ up spins. Counting Bethe states is a notoriously tricky problem. Bethe roots only correspond to $\mathfrak{su}(2)$ highest-weight states [2], while descendants are obtained by sending some roots to infinity [3]. Among the solutions, unphysical ones need to be excluded to obtain the correct eigenstates [4–6].[1]

A physical way to regularize the Bethe roots is to introduce a twisted periodic boundary condition [7]. It is equivalent to turning on a magnetic flux through the closed chain [8], which breaks the $\mathfrak{su}(2)$ degeneracies of the states. Empirically, one finds that all the Bethe roots become physical and account for all the eigenstates in the $M$-magnon sector [9–11]. The bijection between the physical Bethe roots and the eigenstates can be proved by reformulating the BAEs in terms of a set of functional relations [12]. See e.g. [13–15] and references therein. Multi-component models are diagonalized by the nested Bethe ansatz [16,17]. It is expected that turning on multiple twist parameters will make all solutions physical, although a general proof is not known. Models with multiple twist parameters have rich structures, while these models are less explored. The competition of twist parameters results in singular loci where the number of solutions changes. In practice, counting states is still a challenge. Powerful algorithms for solving the BAEs have recently been developed [13, 18, 19]. However, one quickly encounters the curse of dimensionality as the magnon number is increased.

A complementary approach to the solution-counting problem is via gauge theory. The Bethe/Gauge correspondence relates eigenstates of quantum integrable systems with the ground states of 2d supersymmetric theories [20, 21]. The Witten index is a fundamental observable in a supersymmetric theory that counts the vacua [22]. A basic question, still unanswered, is whether physical Bethe states can be correctly counted by the Witten index. The goal of this paper is to provide a definite answer to this problem.

Symmetries of integrable models also provide a complementary way to address questions about supersymmetric vacua. The number of Bethe states changes at special loci in the space of twist parameters. The corresponding picture is that there are singularities in the gauge-theory moduli space where the minimum energy vanishes and some Coulomb vacua may escape to infinity [23]. Little is known about the behavior of gauge theories at singularities. Strictly speaking, the Witten index is well defined away from such singular loci. Nevertheless, the highest-weight property of the Bethe states allows us to predict the number of Coulomb vacua even at singularities in the moduli space, which is difficult to study using field-theory methods. As we will see in concrete examples for the rank-1 and 2 examples, some discrete Coulomb vacua still survive, and this suggests there exist some well-defined effective theories residing on these singular loci.

The Witten index can be evaluated by taking a limit of the elliptic genus [24], which has been computed by supersymmetric localization [25–27]. The elliptic genus receives contributions from a set of poles specified by the Jeffrey-Kirwan prescription [28]. As we will show, they are in one-to-one correspondence with the eigenstates. The index of a U($M$) gauge theory counts *all* the eigenstates in the $M$-particle sector of a quantum integrable model. We

---

[1]Not all the Bethe states correspond to the eigenstates of the Hamiltonian, see, for example, [5,6]. In this paper, we will only count the physical Bethe states.

show that the solution counting of the multi-component models leads to a new combinatorial problem that is a 3d generalization of restricted occupancy.

The index method can be applied to a large class of quantum integrable models. We study the Kondo model and the 1d t-J model with twisted boundary conditions. For the SU(3) spin chain and the t-J model, we identify the singular loci where the number of vacua changes. We propose formulae relating the solution counts for the twisted and untwisted models and test them with an analytic solver developed in [18]. In the untwisted limit, the solution count exactly reproduces the results in the literature.

## 2 The Bethe/gauge correspondence and the Witten index

The Heisenberg spin chain admits an integrable generalization to higher spins, inhomogeneities and a twisted boundary condition [29]. The BAEs take the form

$$\prod_{\alpha=1}^{L}\frac{\lambda_k - \nu_\alpha + is_\alpha}{\lambda_k - \nu_\alpha - is_\alpha} = e^{-t}\prod_{\substack{j=1 \\ j\neq k}}^{M}\frac{\lambda_k - \lambda_j + i}{\lambda_k - \lambda_j - i}\,. \tag{2}$$

According to the Bethe/Gauge dictionary [21], they coincide with the Coulomb vacua equations of a 2d $\mathcal{N} = (2,2)$ supersymmetric gauge theory with gauge group U($M$), $L$ pairs of fundamental and anti-fundamental chiral multiplets $(Q, \widetilde{Q})$, and an adjoint chiral multiplet $\Phi$. The twisted masses for the matter fields correspond to the spins and inhomogeneities. The combination of the Fayet-Iliopoulos parameter and the $\theta$ angle $t = 2\pi\xi + i\theta$ corresponds to the diagonal twist parameter. There is a superpotential of the form

$$\mathcal{W} = \sum_{\alpha=1}^{L} w_\alpha \widetilde{Q}^\alpha \Phi^{2s_\alpha} Q_\alpha\,, \tag{3}$$

where $w_\alpha$ are complex coefficients.

The supersymmetric vacua can be counted by the Witten index [22]:

$$\mathrm{Tr}\,(-1)^F e^{-\beta H}\,. \tag{4}$$

It can be obtained by taking a certain limit of the elliptic genus [24], which is a torus partition function that can be computed via supersymmetric localization [25–27]. Detailed computations are presented in the appendix. The elliptic genus of this model is

$$Z_{T^2} = \sum_{\vec{n}}\prod_{\alpha,\beta=1}^{L}\prod_{m_\alpha=0}^{n_\alpha-1}\frac{\theta_1(\tau|\xi_{\alpha\beta} + (n_\beta - m_\alpha)\lambda - z)}{\theta_1(\tau|\xi_{\alpha\beta} + (n_\beta - m_\alpha)\lambda)}\frac{\theta_1(\tau|-\xi_{\alpha\beta} + (m_\alpha - s_\alpha - s_\beta)\lambda)}{\theta_1(\tau|-\xi_{\alpha\beta} + (m_\alpha - s_\alpha - s_\beta)\lambda + z)}\,. \tag{5}$$

The summation is over all configurations of non-negative integers $\vec{n} = \{n_1, \ldots, n_L\}$ such that $|\vec{n}| := \sum_\alpha n_\alpha = M$. Note that the product vanishes when $m_\alpha = 2s_\alpha$. Therefore, the summation truncates to configurations where $n_\alpha \leq 2s_\alpha$. The Witten index is obtained in the $z \to 0$ limit, where the summand tends to 1. The number of vacua is the possible configurations of $M$ boxes into $L$ columns, each with capacity $2s_\alpha$.

This is a classic combinatorial problem known as restricted occupancy [30, 31], as shown in Fig. 1. Since each column may be filled with $n_\alpha = 0, \ldots, 2s_\alpha$ boxes, it may be identified with a state with spin $s_\alpha$. Each configuration is in one-to-one correspondence with an $M$-magnon state in a spin chain. When all $s_\alpha$ are equal to $s$, the number of choices is given by

$$c_s(L; M) = \sum_{j=0}^{L}(-1)^j\binom{L}{j}\binom{L + M - 1 - (2s + 1)j}{L - 1}\,. \tag{6}$$

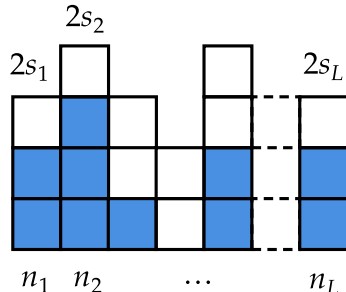

Figure 1: The pole configurations of the elliptic genus give rise to a restricted occupancy problem. Each column with at most $s_\alpha$ blue boxes corresponds to a spin-$s_\alpha$ state at each site of the spin chain. The total number of blue boxes is $M$.

It is independent of the spin for $2s \geq M$, for there may be at most $M$ boxes in a column. We have used numerical methods to test this fact, which is not obvious from the BAEs. Summing $c_s(L; M)$ over $M = 0, \ldots, 2sL$, we count all possible ways of placing $0, \ldots, 2s$ boxes in each column, yielding a total of $(2s + 1)^L$ states.

We remark that a truncation of the partition function is common in theories with polynomial superpotentials in the adjoint fields [32,33]. The same combinatorial problem also arises in [34], where the number of physical Bethe states for the untwisted case is shown to be equal to the difference $c_s(L; M) - c_s(L; M - 1)$, although the twisted case was not considered.

The spin chain has a $\mathbb{Z}_2$ symmetry that reverses each spin as $s_\alpha^z \leftrightarrow -s_\alpha^z$. The total magnetization $S^z = \sum_\alpha s_\alpha - M$ and the magnetic flux $t$ are also reversed. This implies a duality to a U($2\sum_\alpha s_\alpha - M$) gauge theory with the same matter content and superpotential. Such duality has been tested by the sphere partition function, where the parameters are found to transform as expected [32,35]. It may also be tested by an index computation analogous to [33].

## 3 Nested Bethe ansatz and 3d restricted occupancy

Multi-component models are diagonalized by the nested Bethe ansatz [16,17]. The excitations for one component become the pseudo-vacuum for another. For the twisted SU($r + 1$) spin chain, the Bethe roots $\lambda_k^{(a)}, k = 1, \ldots, M_a$ satisfy the equations [36]

$$
\prod_{\alpha=1}^{L} \frac{\lambda_k^{(1)} - \nu_\alpha + is_\alpha}{\lambda_k^{(1)} - \nu_\alpha - is_\alpha} = e^{-t_1} \prod_{\substack{j=1 \\ j \neq k}}^{M_1} \frac{\lambda_k^{(1)} - \lambda_j^{(1)} + i}{\lambda_k^{(1)} - \lambda_j^{(1)} - i} \prod_{j=1}^{M_2} \frac{\lambda_k^{(1)} - \lambda_j^{(2)} - \frac{i}{2}}{\lambda_k^{(1)} - \lambda_j^{(2)} + \frac{i}{2}},
$$

$$
1 = e^{-t_a} \prod_{j=1}^{M_{a-1}} \frac{\lambda_k^{(a)} - \lambda_j^{(a-1)} - \frac{i}{2}}{\lambda_k^{(a)} - \lambda_j^{(a-1)} + \frac{i}{2}} \prod_{\substack{j=1 \\ j \neq k}}^{M_a} \frac{\lambda_k^{(a)} - \lambda_j^{(a)} + i}{\lambda_k^{(a)} - \lambda_j^{(a)} - i} \prod_{j=1}^{M_{a+1}} \frac{\lambda_k^{(a)} - \lambda_j^{(a+1)} - \frac{i}{2}}{\lambda_k^{(a)} - \lambda_j^{(a+1)} + \frac{i}{2}},
\tag{7}
$$

for $a = 2, \cdots, r$. The equations determine eigenstates with $M_a - M_{a+1}$ objects for each component. Their gauge dual is an $A_r$-type linear quiver gauge theory, as shown in Fig. 2.

To keep the notation compact, we consider the case when all $s_\alpha$ are equal. We turn on a superpotential of the form

$$
\mathcal{W} = \text{Tr}\left[ \widetilde{Q} \Phi_1^{2s} Q + \sum_{a=1}^{r-1} \left( \widetilde{B}_a \Phi_{a+1} B_a + B_a \Phi_a \widetilde{B}_a \right) \right],
\tag{8}
$$

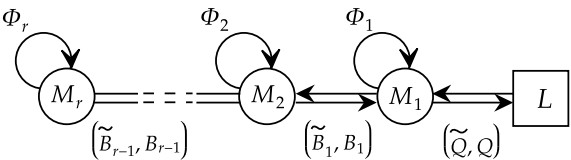

Figure 2: The linear quiver gauge theory corresponding to the SU($r+1$) spin chain.

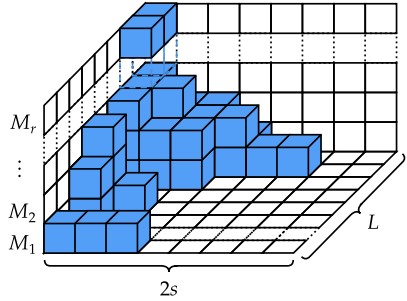

Figure 3: A 3d restricted occupancy problem. On each level we have a 2d restricted occupancy with $n_\alpha^{(a)}$ boxes in each row and a total of $M_a$ boxes. Each row on a higher level has no more boxes than the lower level.

with all coefficients set to unity.

The computation of the Witten index yields a novel 3d restricted occupancy problem, as shown in Fig. 3. We now count all configurations $\vec{n}^{(a)}$ such that $n_\alpha^{(a)} \leq n_\alpha^{(a-1)}$. Intuitively, we count ways of constructing a building on an $L \times 2s$ ground with $M_a$ rooms on each level, from left to right, such that no room on a higher level protrudes over the lower level. We may also view it as a restricted colored partition $[n_\alpha^{(1)}, \ldots, n_\alpha^{(r)}]$ for $\alpha = 1, \ldots, L$. Each partition is restricted to lie inside an $r \times 2s$ rectangle, with the additional constraint $\sum_{\alpha=1}^{L} n_\alpha^{(a)} = M_a$ on each level.

We see the correspondence to an SU($r+1$) spin-$s$ state as follows. Each site corresponds to a $2s$-th symmetric power of the fundamental representation of $\mathfrak{su}(r+1)$, whose weights may be mapped to semi-standard Young tableaux $\square\square\cdots\square$ of length $2s$ with a set of non-increasing integers $0 \leq h_i \leq r$ in the $i$-th box. $h_i$ may be identified with the height of the $i$-th column of the partition $n_\alpha^{(a)}$ for each $\alpha = 1, \ldots, L$. See Fig. 4.

Each partition is defined by a boundary path consisting of $2s$ horizontal and $r$ vertical steps. The total number of configurations is precisely the total number of states in an SU($r+1$) spin-$s$ chain [34]:

$$\sum_{0 \leq M_r \leq \cdots \leq M_1 \leq 2sL} c_s(L; M_1, \ldots, M_r) = \binom{2s+r}{r}^L. \tag{9}$$

We remark that the eigenstates of the untwisted SU($r+1$) spin chain was counted in [37], which leads to another combinatorial problem. It would be an interesting mathematical question to study their relations.

Duality exchanges the pseudo-vaccum with the excitations. It acts locally on a gauge node, corresponding to a Weyl reflection of the associated root [38]. The rank of the gauge group transforms as

$$M_a' = M_{a-1} + M_{a+1} - M_a, \tag{10}$$

where $M_0 := 2sL$ and $M_{r+1} := 0$. In the restricted occupancy picture, the dual configuration is

$$\vec{n}'^{(a)} = \vec{n}^{(a-1)} + \vec{n}^{(a+1)} - \vec{n}^{(a)}. \tag{11}$$

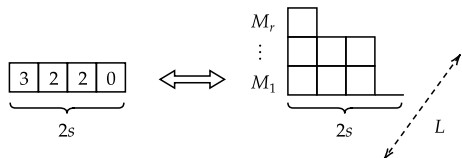

Figure 4: At each site, the $\mathfrak{su}(r+1)$ weight labelled by a semi-standard Young tableau corresponds to a partition in the 3d restricted occupancy problem.

## 4 The untwisted limit

Let us now consider the rank-one case. In the $t \to 0$ limit, the SU(2) symmetry is restored. The physical Bethe states degenerate and organize into $\mathfrak{su}(2)$ highest-weight states of fixed spin with multiplicity determined by the tensor product. The descendants are those states not annihilated by $S^+$. Thus they are in one-to-one correspondence with states with spin $s + 1$, or $M - 1$ magnons. One magnon decouples from the system and flies off to infinity [3]. Another way to see this is that the momentum of one magnon is frozen to zero and decouples from the dynamics. Thus the number of highest-weight states labeled by each set of solutions to the untwisted BAEs is

$$c_s(L; M) - c_s(L; M - 1). \tag{12}$$

In the higher-rank case when one $t_a \to 0$, an SU(2) subgroup of the SU($r + 1$) symmetry is restored. We postulate the following: descendants corresponding to infinite rapidities are removed. There should be

$$c_s(L; M_1, \cdots, M_r) - c_s(L; M_1, \cdots, M_a - 1, \cdots, M_r), \tag{13}$$

solutions to the partially twisted BAEs.

There are other singular loci where the number of solutions change. For the SU(3) chain, we propose a formula for the solution count when $t_1 + t_2 = 0$:

$$c_s(L; M_1, M_2) - c_s(L; M_1 - 1, M_2 - 1). \tag{14}$$

For the fully untwisted case, we propose a formula for the number of highest-weight states:

$$\begin{aligned} &c_s(L; M_1, M_2) - c_s(L; M_1 - 1, M_2) - c_s(L; M_1, M_2 - 1) \\ &+ c_s(L; M_1 - 2, M_2 - 1) + c_s(L; M_1 - 1, M_2 - 2) - c_s(L; M_1 - 2, M_2 - 2). \end{aligned} \tag{15}$$

Our predictions can be tested by explicitly solving the BAEs. We have generalized the analytic solver developed in [18] with twist parameters, which is capable of computing up to a total of $\sim 500$ solutions. Some results are shown in Table 1. For the untwisted case, the prediction agrees with the Littlewood-Richardson coefficient [39] in every case. Proposals for higher rank will be discussed in [40].

Table 1: The solution count for the BAEs for the SU(3) $s = 1/2$ spin chain.

| $(L; M_1, M_2)$ | generic $t$ | $t_1 = 0$ | $t_2 = 0$ | $t_1 + t_2 = 0$ | $t_1, t_2 = 0$ |
|---|---|---|---|---|---|
| (7;4,1) | 140 | 105 | 35 | 105 | 21 |
| (8;4,2) | 420 | 140 | 252 | 252 | 56 |
| (10;4,1) | 840 | 630 | 480 | 720 | 315 |



Figure 5: An $A_2$ quiver for the t-J model.

# 5 Applications to other models

**The Kondo model**  Consider the Kondo model with an impurity of arbitrary spin $s$ [41]. It is solved by the BAEs (2) with $s_\alpha = 1/2$ for $\alpha = 1,\ldots,L$ and $s_{L+1} = s$. For non-zero $t$, we count $(M-i)$-magnon states when the impurity has spin $s-i$ for $i = 0,\ldots,2s$:

$$c_s^{\text{K}}(L;M) = \sum_{i=0}^{2s} \binom{L}{M-i}.$$ (16)

The total number of states is $2^L(2s+1)$, as expected. In the $t \to 0$ limit, the SU(2) symmetry is restored and we recover the solution count of the untwisted model [41]

$$c_s^{\text{K}}(L;M) - c_s^{\text{K}}(L;M-1) = \binom{L}{M} - \binom{L}{M-2s-1}.$$ (17)

**The t-J model**  The 1d t-J model describes the spin-hopping interaction of $M_1$ electrons in a lattice of $L$ sites with $M_2$ spin-down excitations [42]. The Hamiltonian can be diagonalized by the BAEs corresponding to the superalgebra $\mathfrak{sl}(1|2)$ [43]. There are three equivalent BAEs [44], associated with choices of Borel subalgebras. This leads to dualities between $A_2$ quiver gauge theories [45]; one of which is shown in Fig. 5.

Turning on a $\widetilde{Q}\Phi^{2s}Q$-type superpotential for the adjoint field leads naturally to a spin-$s$ generalization of the t-J model where each electron carries spin $s$. Note that there is an adjoint field only for the $M_2$ node. The elliptic genus receives contributions from the poles at

$$\begin{aligned}
\left(u_i^{(1)}\right)_* &= \xi_\alpha - \chi^{(1)}, \\
\left(u_i^{(2)}\right)_* &= \xi_\beta - \chi^{(1)} - \chi^{(2)} - m_\beta \lambda.
\end{aligned}$$ (18)

Here $\alpha \in I_1$ and $\beta \in I_2$, where $I_1$ is a choice of $M_1$ integers from $\{1,\ldots,L\}$ and $I_2$ is a choice of $M_2$ integers from $I_1$. Counting the poles, we obtain the solution count for the twisted spin-$s$ t-J model:

$$c_s^{\text{tJ}}(L;M_1,M_2) = \binom{L}{M_1} c_s(M_1;M_2).$$ (19)

Since each site may be either vacant or a spin-$s$ state, we find a total of $(2s+2)^L$ states by summing over all $0 \le M_2 \le M_1 \le L$.

In the untwisted limit, we conjecture the solution count to be

$$\sum_{i=0}^{\infty} (-1)^i \left[ c_s^{\text{tJ}}(L;M_1,M_2-i) - c_s^{\text{tJ}}(L;M_1-i-1,M_2-i) \right].$$ (20)

For $s = 1/2$, we recover the solution count [46]

$$\frac{L!(L-2M_1+M_2+1)}{M_1 M_2!(L-M_1)!(M_1-M_2-1)!(L-M_1+M_2+1)}.$$ (21)

A derivation of (20), based on characters of Lie superalgebras, will be presented in [40].

# 6 Discussion

It would be interesting to extend the index techniques to important examples such as the XXZ spin chain, the Hubbard model, and other superspin chains [47–49]. We expect that the physical states of an XXZ spin chain can be similarly counted by the 3d Witten index. One important caveat is that when the anisotropy parameter is a root of unity, there may be continuous solutions to the BAEs [50,51] that correspond to non-compact Coulomb branches in the gauge theory. Examples of such non-compact Coulomb branches in a related context were studied recently in [52]. In this paper, we presented the index of an $A_2$ quiver for the t-J model. A different spin-$s$ realization was studied in [53], which would predict new gauge-theory dualities. For $s = 1/2$, such dualities were realized in string theory [45] and then extended to $\mathfrak{gl}(m|n)$ spin chains [49]. It would be interesting to consider higher $s$.

It would also be interesting to consider open chains [54] and general boundary conditions. The form of the off-diagonal BAEs [55] suggests that the existing Bethe/Gauge dictionary needs to be expanded. In this paper, we have obtained the solution count from the leading $z \to 0$ part of the elliptic genus. It would be interesting to extract other useful information from the elliptic genus. For example, the $\chi_y$ genus receives contributions from the Higgs branch.

The limitation of the index is that it does not know about the detailed nature of the vacua. A common assumption on Coulomb vacua is that repeated solutions $\lambda_j = \lambda_k$ should be excluded because they correspond to strongly-coupled regions. Solutions at $\lambda_k = v_\alpha \pm i s_\alpha$ should also be excluded because quantum Higgs branches may emanate from there [56,57]. The rôle of Higgsing was elucidated in [38,58]. Despite evidence for the absence of vacua at such points, a proof is not known. Analysis of the BAEs show, however, that some such solutions are physical in the untwisted model [4,6]. This hints at new vacua at the singular loci and calls for a more refined solution counting, using the Witten index and other methods.

# Acknowledgments

We thank Jin Chen, Jie Gu, Wei Gu, Yunfeng Jiang, Nicolai Reshetikhin, Ruijie Xu, Wen-Li Yang, and Yang Zhang for discussions. The work of H. S. is supported in part by the Beijing Postdoctoral Research Foundation. P. Z. is supported in part by a China Postdoctoral Science Foundation Special Fund, grant number Y9Y2231. R. Z. is supported by the National Natural Science Foundation of China under Grant No. 12105198 and the High-level personnel project of Jiangsu Province (JSSCBS20210709). H. Z. is supported by the China Postdoctoral Science Foundation with grant No. 2022M720509.

# A Computation of the elliptic genus

We follow the convention in [26,27] for the elliptic genus. For a 2d $\mathcal{N} = (2,2)$ supersymmetric theory, the elliptic genus is a refined Witten index defined as

$$Z_{T^2} = \mathrm{Tr}\,(-1)^F q^{H_L} \bar{q}^{H_R} y^J \prod_i x_i^{K_i}. \tag{A.1}$$

Here $F$ is the fermion number, $q = \exp 2\pi i \tau$ is defined by the modular parameter of the torus, $H_L$ and $H_R$ are the left- and right-moving Hamiltonians, respectively. $J$ is the charge of the left-moving U(1) R symmetry and $K_i$ are the charges of the flavor symmetry. The global symmetry fugacities $y$ and $x_i$ are related to the holonomies of the background gauge fields as

$$y = e^{2\pi i z}, \qquad x_i = e^{2\pi i u_i}. \tag{A.2}$$

The supersymmetric localization formula gives [27]

$$Z_{T^2} = \frac{1}{|W|} \sum_{u_* \in \mathfrak{M}^*_{\text{sing}}} \text{JK-Res}(Q(u_*), \eta) \, Z_{\text{1-loop}} \,. \tag{A.3}$$

Here $|W|$ is the order of the Weyl group and $\mathfrak{M}^*_{\text{sing}}$ is the set of poles that contribute to the Jeffrey-Kirwan (JK) residue, defined as follows: $Q(u_*)$ is the subset of charges that lie in the chamber defined by the stability parameter $\eta$. The result is independent of the choice of $\eta$. The one-loop determinant depends on the field content of the gauge theory. The contribution of a vector multiplet is

$$Z_V = \left[ \frac{2\pi \eta(q)^3}{\theta_1(\tau|-z)} \right]^{\text{rk}\,G} \prod_{\alpha \in \Delta} \frac{\theta_1(\tau|\alpha \cdot u)}{\theta_1(\tau|\alpha \cdot u - z)} \prod_{i=1}^{\text{rk}\,G} du_i \,. \tag{A.4}$$

The contribution of a chiral multiplet is

$$Z_C = \prod_{\rho \in \mathfrak{R}} \frac{\theta_1(\tau|\rho \cdot u + (J-1)z)}{\theta_1(\tau|\rho \cdot u + Jz)} \,. \tag{A.5}$$

The product is over all the weights $\rho$ of the representation $\mathfrak{R}$ of the gauge and flavor groups.

## A.1 Rank one

The flavor symmetry is $U(L)_Q \times U(L)_{\widetilde{Q}} \times U(1)_\Phi$. Due to the superpotential (3), each $U(L)$ is broken to $U(1)^L$, which rotates the individual components. We further rewrite $U(1)_Q \times U(1)_{\widetilde{Q}}$ into the vector and axial parts $U(1)_{\text{vec}} \times U(1)_{\text{axi}}$. Note that the R charge of a chiral multiplet enters in its one-loop determinant (A.5), which is cumbersome in calculations. It is convenient to redefine the flavor symmetry by mixing with the left-moving R symmetry $U(1)_J$ as $U(1)'_{\text{vec}} := U(1)_{\text{vec}} + U(1)_J$, etc. One may introduce the holonomies $\chi_\alpha$, $\xi_\alpha$, and $\lambda$ for $U(1)'_{\text{vec}}$, $U(1)'_{\text{axi}}$, and $U(1)'_\Phi$, respectively. Since the superpotential is invariant under the original flavor symmetry and charged 1 under the left-moving R symmetry, this leads to the constraint

$$2\chi_\alpha + 2s_\alpha \lambda = z \,. \tag{A.6}$$

The one-loop determinant of this gauge theory is, up to a possible sign,

$$\begin{aligned}
Z_{\text{1-loop}} = & \left[ \frac{2\pi \eta(q)^3}{\theta_1(\tau|-z)} \right]^M \left[ \prod_{i \neq j}^M \frac{\theta_1(\tau|u_{ij})}{\theta_1(\tau|u_{ij}-z)} \right] \left[ \prod_{i,j=1}^M \frac{\theta_1(\tau|u_{ij}+\lambda-z)}{\theta_1(\tau|u_{ij}+\lambda)} \right] \\
& \times \prod_{i=1}^M \prod_{\alpha=1}^L \frac{\theta_1(\tau|u_i - \xi_\alpha + \chi_\alpha - z)}{\theta_1(\tau|u_i - \xi_\alpha + \chi_\alpha)} \frac{\theta_1(\tau|-u_i + \xi_\alpha + \chi_\alpha - z)}{\theta_1(\tau|-u_i + \xi_\alpha + \chi_\alpha)} \, d^M u \,.
\end{aligned} \tag{A.7}$$

Here $u_{ij} := u_i - u_j$. Taking $\eta = (1, \ldots, 1)$, the poles that contribute are located at $u_i = u_j + z$, $u_i = u_j - \lambda$, and $u_i = \xi_\alpha - \chi_\alpha$. The first type of poles leads to a vanishing residue. Thus the non-trivial poles form a tower of the form

$$(u_i)_* = \xi_\alpha - \chi_\alpha - m_\alpha \lambda \,, \tag{A.8}$$

for $m_\alpha = 0, \ldots, n_\alpha - 1$. The poles are labeled by possible configurations of $\vec{n} = \{n_1, \ldots, n_L\}$ with $n_\alpha \geq 0$ and $\sum_\alpha n_\alpha = M$. The JK residue integral gives [25, 27]

$$Z_{T^2} = \sum_{\vec{n}} \prod_{\alpha,\beta=1}^L \prod_{m_\alpha=0}^{n_\alpha-1} \frac{\theta_1(\tau|\xi_{\alpha\beta} + (n_\beta - m_\alpha)\lambda - z)}{\theta_1(\tau|\xi_{\alpha\beta} + (n_\beta - m_\alpha)\lambda)} \frac{\theta_1(\tau|-\xi_{\alpha\beta} + \chi_\alpha + \chi_\beta - z + m_\alpha\lambda)}{\theta_1(\tau|-\xi_{\alpha\beta} + \chi_\alpha + \chi_\beta + m_\alpha\lambda)} \,. \tag{A.9}$$

The products due to the vector multiplet telescope with the adjoint chiral multiplet, which then cancel with the fundamental chiral multiplet to yield the first line. The second line is due to the anti-fundamental chiral multiplet. Imposing the holonomy constraints (A.6) then leads to (5).

## A.2 Higher rank

For the linear quiver gauge theory defined in Section 3, the flavor symmetry preserved by the superpotential (8) is $U(1)_{vec} \times SU(L)_{axi}$ for $(Q, \widetilde{Q})$, $U(1)_{vec} \times U(1)_{axi}$ for each $(B_a, \widetilde{B}_a)$, and $U(1)_{\Phi_a}$ for each $\Phi_a$. We mix the flavor symmetry with the R symmetry and introduce the holonomies $\chi^{(a)}$ for $U(1)'_{vec}$, $\xi_\alpha$ for $SU(L)'_{axi}$, and $\lambda^{(a)}$ for $U(1)'_{\Phi_a}$. The superpotential imposes the following constraints:

$$
\begin{aligned}
2\chi^{(1)} + 2s\lambda^{(1)} = 2\chi^{(2)} + \lambda^{(2)} = \cdots = 2\chi^{(r)} + \lambda^{(r)} = z, \\
2\chi^{(2)} + \lambda^{(1)} = \cdots = 2\chi^{(r)} + \lambda^{(r-1)} = z,
\end{aligned}
\tag{A.10}
$$

which imply that all $\lambda^{(a)}$ are equal and $\chi^{(2)} = \cdots = \chi^{(r)}$.

The one-loop determinant of the gauge theory can be read out by multiplying all the matter contributions

$$
Z_{1\text{-loop}} = \prod_{a=1}^{r} Z_{V_a} Z_{\Phi_a} Z_{B_{a-1}} Z_{\widetilde{B}_{a-1}},
\tag{A.11}
$$

where $(B_0, \widetilde{B}_0) := (Q, \widetilde{Q})$. The JK prescription picks up non-trivial contributions from the poles at

$$
\begin{aligned}
u_i^{(a)} - u_j^{(a)} + \lambda = 0, \\
u_i^{(a)} - u_i^{(a-1)} + \chi^{(a)} = 0.
\end{aligned}
\tag{A.12}
$$

for $a = 1, \ldots, r$ and $u_i^{(0)} := \xi_i$. We label the poles by

$$
\left( u_i^{(a)} \right)_* = \xi_\alpha - \sum_{i=1}^{a} \chi^{(i)} - m_\alpha^{(a)} \lambda,
\tag{A.13}
$$

where $m_\alpha^{(a)} = 0, \ldots, n_\alpha^{(a)} - 1$. We evaluate the JK residue as before. Applying the holonomy constraints (A.10) and the identity $\theta_1(\tau | -z) = -\theta_1(\tau | z)$, we find

$$
Z_{T^2} = \sum_{\{\vec{n}^{(a)}\}} \prod_{\alpha,\beta=1}^{L} \prod_{m_\alpha^{(a)}=0}^{n_\alpha^{(a)}-1} \frac{\theta_1(\tau | \xi_{\alpha\beta} + (n_\beta^{(a)} - m_\alpha^{(a)})\lambda - z)}{\theta_1(\tau | \xi_{\alpha\beta} + (n_\beta^{(a)} - m_\alpha^{(a)})\lambda)} \frac{\theta_1(\tau | \xi_{\alpha\beta} + (n_\beta^{(a-1)} - m_\alpha^{(a)})\lambda)}{\theta_1(\tau | \xi_{\alpha\beta} + (n_\beta^{(a-1)} - m_\alpha^{(a)})\lambda - z)}.
\tag{A.14}
$$

The summation is over all configurations subject to $|\vec{n}^{(a)}| = M_a$ and $n_\alpha^{(0)} := 2s$. The products from the bi-fundamental chiral multiplets $Z_{B_{a-1}} Z_{\widetilde{B}_{a-1}}$ telescope and cancel against the extra factor from $Z_{V_a} Z_{\Phi_a}$ to yield the second line of (A.14). The product is truncated to configurations where

$$
n_\alpha^{(a)} \le n_\alpha^{(a-1)}.
\tag{A.15}
$$

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
