# Peer review of "Bethe-State Counting and Witten Index"

_SciPost Physics, doi:SciPost Phys. 15, 103 (2023)_

## Round 1 · Referee Report · Anonymous (Referee 1) · 2023-4-16

Strengths

1-Novelty
2-Rigour

Weaknesses

N/A

Report

This paper contains a very interesting study of Bethe-eigenstate counting via the correspondence with the vacua of 2D supersymmetric gauge theories, the latter being enumerated by the Witten index. In this way, the authors can provide an alternative approach to counting the number of solutions of a vast array of different systems of Bethe equations, by reducing the problem to the evaluation of the Witten index in the corresponding 2D gauge theories. The approach is alternative to the brute force counting of solutions which in several cases is available in the literature - counting which is then used to validate the results obtained via the supersymmetric-gauge-theory route.

The paper appears solid and the results are definitely interesting. I wish to express my recommendation for publication.

I would like to add a few considerations which the authors may or may not wish to expand upon. It seems to me that in a number of cases, which the authors analyse in detail, the degeneracy is entirely removed, and the most stringent test of the new counting method seems to be represented by reproducing the total dimension of the Hilbert space. Whilst I agree that this is an important benchmark, I fail to find it particularly surprising or exciting: a number like (2s+1)^L can be anticipated without the help of much creativity. Perhaps the authors could stress a bit more why this is non-trivial and, most importantly, what we really learn from the rephrasing / breaking down of this number via the supersymmetric-gauge-theory route.

On the contrary, the cases with degeneracy appear to my mind to be far more interesting. In this case though the authors provide some excellent conjectures which however are not very extensively motivated, and are often simply stated with only a short phrase to accompany them. I think that it is quite impressive that they match the known results, nevertheless it would be interesting to perhaps be given a bit more of an insight of how formulae such as (14) or (20) have been arrived at.

Requested changes

N/A

  • validity: high
  • significance: high
  • originality: high
  • clarity: good
  • formatting: perfect
  • grammar: excellent

Author:  Hao Zou  on 2023-06-19  [id 3743]

(in reply to Report 1 on 2023-04-16)
Category:
remark

We would like to thank you for your comments. The degeneracy case is indeed very interesting. As a matter of fact, we have a conjecture to obtain the equation (14) or (20) and a rigorous proof is working in progress based on representation theory.

---

## Round 1 · Referee Report · Anonymous (Referee 2) · 2023-5-26

Strengths

Powerfull new approach
Numeric checks

Weaknesses

The subtleties related to physical/unphysical solutions could be clarified a bit

Report

Context

Various integrable quantum systems have been studied for decades, and Bethe equations (BAE) have proven to encode key properties of these models.

Counting the number of solutions of Bethe equations has always been a difficult task, which is all the more important since it allows to deduce the completeness of the Bethe Ansatz. By studying how to count (and to characterize) solutions of Bethe equations, people have noticed we should actually not aim at counting all solutions to Bethe equations (1) and (7), but only the "physical solutions". According to some authors this means that the formulation (1) and (7) of the Bethe Equations should be replaced by an other formulation (as a functionnal relation) which doesn't have unphysical solutions.

Achievements of this paper

This paper gives a very promising approach to count the solutions, based on Bethe-gauge correspondance. This approach is detailed for $A_n$ type spin chains in this paper, but the method should be very general, and may not only allow to count Bethe states.

With this method, the counting of solutions reduces to another combinatorial problem, the occupancy problem (which has a suprisingly simple formulation). The author use it to count the number of Bethe states, and compare it with numerical solutions of Bethe equations. They do it for various degeneracies of this twist, which is know to be a real challenge when people study BAEs.

I do recommand this paper for publication, due to the quality of its results and the clarity of its explainations.

Remark

I however have one remark that the author may be able to reply and which may lead them to wish to add a few remarks in their text:

As explained in the introduction of this report, the number of solutions to Bethe equations depends on the formulation which is chosen for the BAE. The formulation (1) and (7) of the BAE (as aopposed to some "Wronskian" formulations) admits non-physical solutions. I assume that what the authors count is not the total number of solutions of these BAE, but only the number of physical solutions. If this is the case indeed, I think the authors could either insist a bit more on this fact in the main text (and explain what is meant by "physical" vs "unphysical" solutions), or write another formulation of BAEs in order to indeed obtain, that what they are counting is the number of solutions of the BAE (with this appropriate formulation of BAE).

Requested changes

1- The authors may wish to clarify whether they count "the number of solutions to BAE" or "the number of 'physical' solutions to BAE"

  • validity: -
  • significance: -
  • originality: -
  • clarity: -
  • formatting: -
  • grammar: -

Author:  Hao Zou  on 2023-06-19  [id 3742]

(in reply to Report 2 on 2023-05-26)
Category:
correction

We would like to thank the referee for the comments and suggestions. We agree that the Bethe state is more general and we indeed only count the physical ones. We have made several changes, adding “physical” in front of “Bethe states.” In particular, we have added a footnote to address this point at the end of the first paragraph of the introduction section. Please see version 2 of our submission.

---

## Round 2 · List of Changes

1. In the abstract: “We count the Bethe states … ” to “We count the physical Bethe states …”
  2. At the end of the first paragraph of Introduction: we added a footnote [7]: “Not all the Bethe states correspond to the eigenstates of the Hamiltonian, see, for example, [5, 6]. In this paper, we will only count the physical Bethe states. ”
  3. In paragraph starting with “A complementary approach …” in the Introduction: we changed “whether Bethe states can be …” to “whether physical Bethe states can be … ”
  4. In the paragraph below Equation (9): we changed “We remark that the Bethe states of …” to “We remark that the eigenstates of … ”
  5. In the first paragraph of Section IV: we added a word “physical” in the second sentence.

---

## Editorial Decision

published